# Intelligent Security Model for Password Generation and Estimation Using Hand Gesture Features

Bashar Saadoon Mahdi [ID], Mustafa Jasim Hadi [ID] and Ayad Rodhan Abbas *[ID]

Department of Computer Science, University of Technology-Iraq, Baghdad 10066, Iraq
* Correspondence: ayad.r.abbas@uotechnology.edu.iq

**Abstract:** Computer security depends mainly on passwords to protect human users from attackers. Therefore, manual and alphanumerical passwords are the most frequent type of computer authentication. However, creating these passwords has significant drawbacks. For example, users often tend to choose passwords based on personal information so that they can be memorable and therefore weak and guessable. In contrast, it is often difficult to remember if the password is difficult to guess. We propose an intelligent security model for password generation and estimation to address these problems using the ensemble learning approach and hand gesture features. This paper proposes two intelligent stages: the first is the password generation stage based on the ensemble learning approach and the proposed S-Box. The second is the password strength estimation stage, also based on the ensemble learning approach. Four well-known classifiers are used: Multi-Layer Perceptron (MLP), Support Vector Machine (SVM), Random Forest Tree (RFT), and AdaBoost applied on two datasets: MNIST images dataset and password strength dataset. The experimental results showed that the hand gesture and password strength classification processes accurately performed at 99% in AUC, Accuracy, F1-measures, Precision, and Recall. As a result, the extracted features of hand gestures will directly impact the complexity of generated passwords, which are very strong, hard to guess, and memorable.

**Keywords:** password generation; password strength estimation; hand gesture recognition; ensemble learning

## 1. Introduction

Security is now a crucial issue in maintaining our accounts in all fields. In most computer security scenarios, user authentication is a critical component. Computer security depends mainly on passwords to protect human users from attackers. Different types of passwords are used for security reasons, but each has drawbacks such as brute-force attacks and shoulder surfing attacks. For example, users often tend to choose text passwords based on personal information so that they can be memorable and, therefore, weak and guessable. On the other hand, a good security password should be memorable for the user and difficult to guess for others [1]. Four other user authentication methods have been used to secure modern systems: Multi-factor authentication, Certificate-based authentication, Biometric authentication, and Token-based authentication [2]. In the same conditions as text-based passwords, graphical passwords can be an alternative to authentication techniques that provide better usability and security. The idea behind graphical passwords is that pictures are easier to remember than letters. Therefore, pictures or patterns are used as passwords instead of letters or numbers, making them easier to remember than text-based passwords [3].

However, these methods require a long time to enter a password. In addition, they may have other issues, such as communication costs for collecting, storing, and transmitting photograph data necessary for system construction and operation [4]. Furthermore, some of these methods do not estimate the strength of generated passwords, and others rely on statistical methods of checking the strength of generated passwords.

This paper proposes an intelligent security model for password generation and estimation using an ensemble learning approach and hand gesture features. These biometric features are extracted from hand gestures and embedded into new passwords.

In this context, the main contributions are as follows:

1.  Information-gain-based feature selection is used to reduce the feature size of the MNIST dataset from 784 to 60 features.
2.  This paper devises an effective hand gesture recognition using an ensemble learning approach to contribute to the process of generating very strong, hard-to-break, and memorable passwords.
3.  We apply the sampling techniques to the password strength dataset to deal with an imbalanced class.
4.  Four well-known classifiers (MLP, SVM, RFT, and AdaBoost) are trained to evaluate password strength. It draws a test password similarity with most of the dataset's weak, medium, and very strong passwords.
5.  We extract the most important features such as password diversity and entropy from the password strength dataset to improve the accuracy of classifiers.
6.  The proposed mechanism makes it easier for the user to create strong and memorable passwords and also provides a mechanism for checking passwords at the same time. Compared with previous work, we find that the system trains hand gestures and passwords with high accuracy.

The remainder of this paper is organized as follows. Section 2 discusses the related works. In Section 3, the MNIST and password strength datasets are presented. Section 4 discusses new and promising methods for generating and estimating strong passwords. Section 5 illustrates the results and comparison with the state-of-the-art. Finally, Section 6 concludes and briefs the scope for future work.

## 2. Related Work

In this section, we briefly present previous related work about two issues: password generation and password strength estimation.

### 2.1. Password Generation

A password generator is a tool that automatically creates secure passwords that are somewhat impossible to crack on user devices using either rule-based, machine learning, or deep learning. The authors in [5] presented a password generator that uses knowledge from the training phase to reduce the time required and find a correct password. They also described the creation of the Fitcrack tool's password generation. Fitcrack is a password recovery program that decrypts encrypted files and retrieves passwords. Their main goal is to add another password generator to the Fitcrack program. When a password is cracked, the generator generates a hashed password that is compared to the hash from the encrypted file. Unfortunately, the rule-based methods continuously expand the size of dictionaries based on permutations, and concatenation is the traditional way to construct password dictionaries. Therefore, the authors [6] presented a novel password-cracking tool based on a machine learning algorithm that splits each training password into meaningful segments using the N-gram Markov model. After that, it learns patterns from the password segments using the Bayesian model. Finally, it generates personalized high-efficiency password dictionaries based on the learned patterns. The experiments showed that the proposed tool outperforms classic rule-based and alphanumeric-pattern-based tools. Deep learning algorithms generate password dictionaries to improve password guessing efficiency. GENPass is deep learning that learns features from various datasets and uses adversarial generation to verify that the output wordlist is accurate in various data types. GENPass's password generator combines probabilistic context-free grammars with LSTM (PL), a recurrent neural network [7]. Another method, called random character utilization with hashing (RCUH), was proposed to generate new passwords while considering user parameters. The proposed model presents a new framework for creating a password by considering

roughly ten user parameters and analyzing the time to crack the produced password to determine the system's strength [8]. In addition, the authors [9] presented an ensemble method that includes classification and guessing methods. They began with a bi-directional generative adversarial-network-based approach for creating individualized passwords with a higher convergence rate. It generates the same number of samples in less time than GAN. Finally, the one-class SVM is trained on the leaked and created passwords to evaluate password strength.

All previous studies offered acceptable results in terms of password complexity. However, it is quite difficult to remember these strong passwords. In addition, they did not consider biometric features extracted from humans' sign language and embedded these features into new passwords. These features will directly impact the complexity of new passwords, which are very strong, hard to break, and memorable.

### 2.2. Password Strength Estimation

The term "password strength" refers to the ability of a password to withstand external attacks. Password cracking and brute-force attacks are examples of external attacks.

There are three primary approaches to estimating password strength. The first traditional approach uses the password's complexity to determine its strength, such as using the Shannon entropy or statistical methods based on certain simple conventions such as the password length and the sort of symbols or characters used (e.g., uppercase, lowercase, or digits) [10–15]. However, many studies have proven that the password-entropy metric is only useful for analyzing the strength of randomly generated passwords, not for gauging the strength of user-chosen passwords [16,17]. The second approach is rule-based password strength, which assesses whether the user's password complies with the service provider's password construction rules, for instance, whether the password is lengthy enough and whether the number of special characters is sufficient. The password measure algorithm breaks a given password into many patterns and then calculates the entropy of each pattern individual [18,19]. Rule-based password strength methods have the advantage of being easy to implement. At the same time, the drawback is that the result of the password strength estimate is not accurate. Therefore, the third recent approach is a machine-learning-based password strength checking treated as a classification task. Many well-known machine learning algorithms are used, such as Artificial Neural Networks (ANN), Logistic regression (LR), Decision tree (DT), Naïve Bayes (NB), and RFT. These predicted models indicate to users whether a password they choose is strong or weak based on the password strength dataset [20,21].

Unfortunately, these studies depend on an imbalanced strength dataset. The accuracy of these models is biased toward strong passwords, at least 74.31%, according to the distribution of class values as shown in Figure 2. Another drawback is that important password features are neglected, such as diversity features and the number of characters, special characters, and digital numbers.

### 2.3. The Applications of Intelligent Data Security

The number of theft cases has been alarmingly increasing in today's globalized and competitive society. Is there a way out of this? Yes, smart home systems would improve the security and safety of our life. It utilizes the idea of one-time password generation and is based on the Internet of Things (IoT). For example, deep-learning password crackers for IoT improved PassGAN's performance, and created two methods for better password cracking: the first involved switching the cost function from an improved Wasserstein GAN based on a CNN to one based on an RNN, and the second involved using the dual-discriminator GAN [22]. Ref. [23] presented an IoT-based smart office area monitoring and control by the implementation of the GSM-based security control panel with one-time password (OTP) creation for the RFID-based smart office monitoring. Ref. [24] also presented a secure and stable OTP generation using fingerprints. To increase the security of digital door locks, Ref. [25] suggested an OTP-based IoT door-lock solution. The system includes a door lock

with many functions such as OTP password generation, live streaming image storage, and lock remote control, as well as a smartphone application with other features. A dialect password recognition system based on CNN was constructed to detect dialect password recognition in smart homes [26]. Ref. [27] presented a self-generated password protection system for an Arduino-Uno-based real-time home security system. This security system's primary task is to detect human presence and notify the user whenever it is essential by sending a text message to the user's previously registered mobile phone number. There are some technical studies on intelligent algorithms and data security: To solve the difficult problem of identifying computers and humans apart, Ref. [28] offered a challenge–response-password-based authentication system based on the Completely Automated Public Turing test. Ref. [29] also offered a concise but comprehensive overview of the data security and privacy protection concerns related to cloud computing across all data life cycle stages. Ref. [30] presented an artificial intelligence algorithm for data-improved encryption utilized in IoT ends and intermediate nodes.

## 3. Datasets

### 3.1. Sign Language Dataset

The original MNIST images dataset is a famous benchmark for image-based machine learning algorithms [31], as shown in Figure 1. Still, researchers have attempted to update it and build drop-in replacements that are more difficult for computer vision and original for real-world applications. For example, most pairs of MNIST digits (784 total pixels per sample) can be discriminated against quite effectively by just one pixel, as stated in a new replacement named the Fashion-MNIST dataset. The sign language MNIST is offered to encourage the community to produce more drop-in replacements. It uses the same CSV format as the other MNISTs, with labels and pixel values in single rows. It is a multi-class challenge with 24 classes of hand motions in the American Sign Language (ASL) letter database (from A to Y or 0 to 24) (except for J and Z, which need motion).

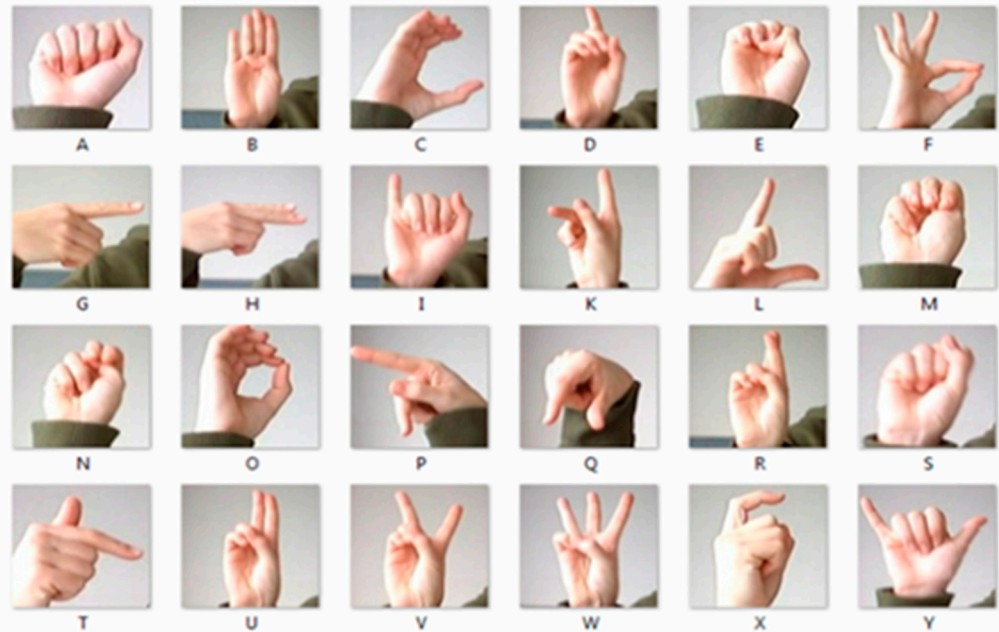

**Figure 1.** Sign language dataset [31].

### 3.2. Password Strength Dataset

The passwords used in this paper are collected from the Kaggle dataset (bansal, 2019). It consists of 669879 passwords of varying strength labeled as either weak, medium, or strong (0, 1, or 2). In addition, some rules determine the password's strength, such as containing digits and special symbols. As shown in Figure 2, the strength class values are

an imbalanced dataset; the percentage distributions of class values are 13.37%, 74.31%, and 12.33% for weak, strong, and medium, respectively.

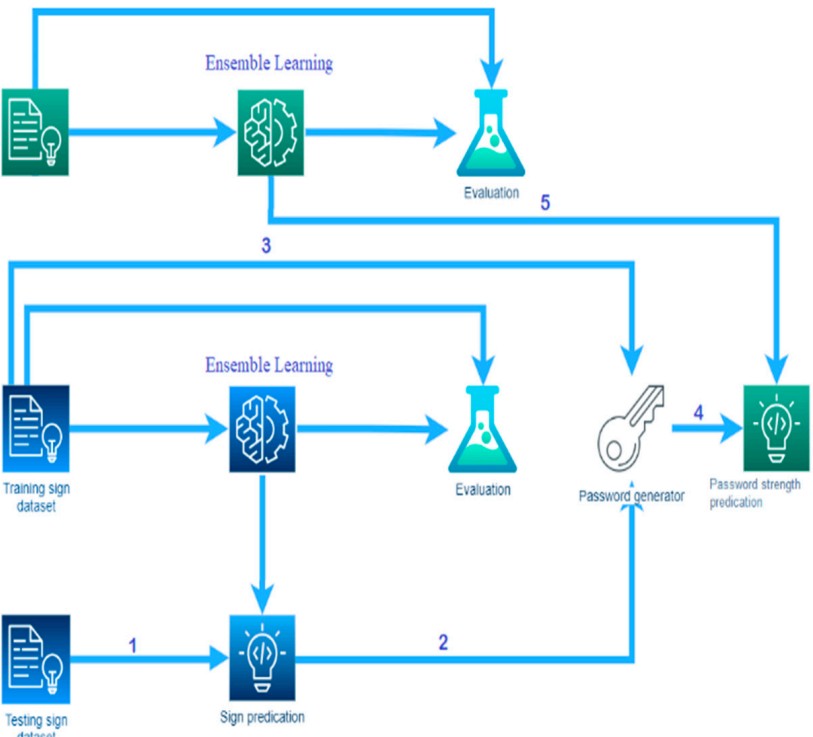

**Figure 2.** Percentage distributions of password strength [32].

### 4. Methods

As shown in Figure 3, this paper presents a new and promising approach for generating and estimating strong and memorable passwords by creating and integrating two intelligent models as follows:

1.  A hand gesture recognition model using an ensemble learning approach to contribute to the process of generating passwords.
2.  A password strength checking model using an ensemble learning approach to estimate the strength of passwords.

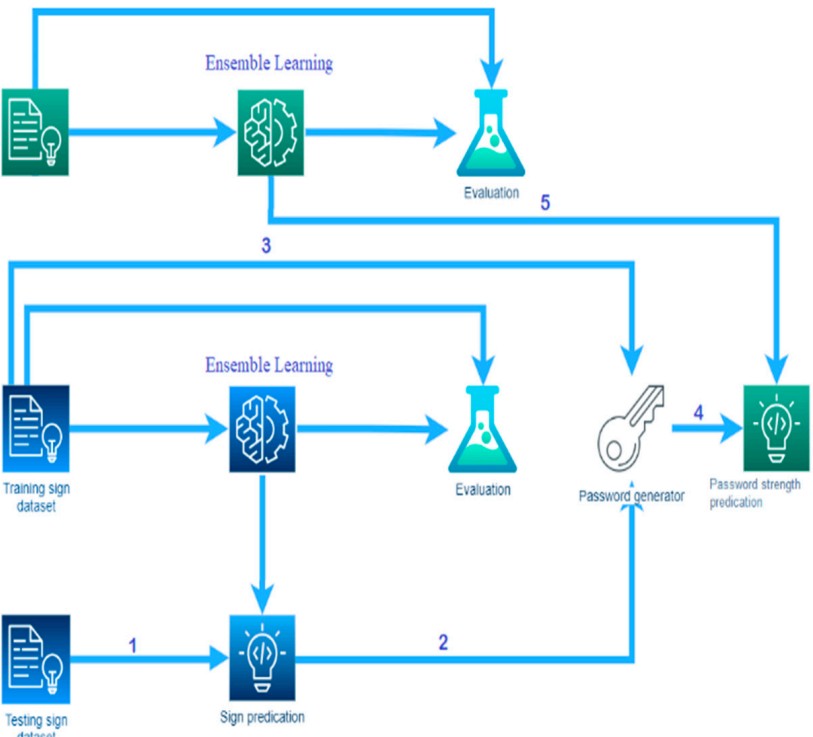

**Figure 3.** The block diagram of the password generation and estimation.

The processing steps of the proposed approach are illustrated as follows:

1. The user tries to choose at least four different signs through hand gestures.
2. After the hand gesture classification, the sign prediction process tries to predict the label for each user's hand motion.
3. Similar features to predicted motion will be retrieved from the training MNIST images dataset and then passed to the password generator.
4. The password generator generates a new password depending on each user's motion features. The password generator has two possible inputs: the label for each user's hand motion and similar features to predicted motion.
5. After the classification of password strength, a new password is passed to the prediction process of the password to estimate its strength. If the new password's strength is either medium or weak, the system will reject it and generate a new password; otherwise, the proposed approach will accept it.

The evaluation processes for each model are used to estimate the whole system's performance. All experiments are carried out on a computer with a 2.5 GHz Core i5 CPU, 8 GB of RAM, and Windows 10 Professional operating system.

### 4.1. The Proposed Password Generation Using Ensemble Learning

This suggested approach intends to build an efficient security model for password generation using hand gestures. This model can be defined into four main processes: feature selection, hand gesture classifiers, hand gesture recognition, and password generation. There are two main reasons to apply an ensemble model over a single model. Performance: Compared to a single contributing model, an ensemble can predict events more accurately. Generalization: The ensemble model's capability to adapt to new datasets. Figure 4 shows the architecture of hand gesture classification using ensemble learning.

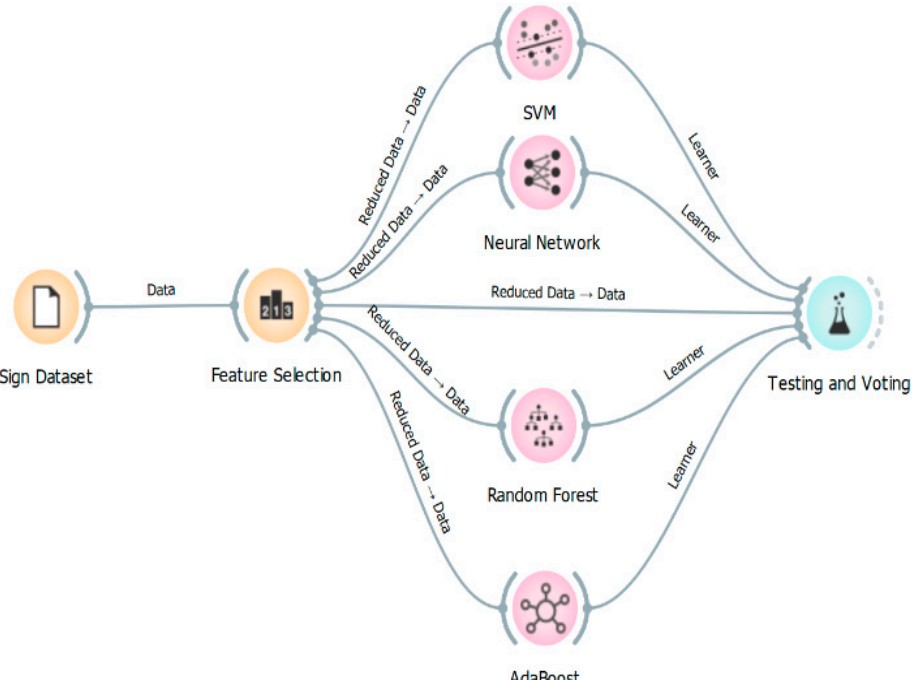

**Figure 4.** The architecture of hand gestures classification using ensemble learning.

### 4.1.1. Feature Selection Method

The major problem of the original MNIST images dataset consists of a high-dimensional feature space with 784 total pixels per sample. In other words, many relevant pixels have a low correlation or are uncorrelated with other pixels and have a high correlation degree with the class of hand gestures. Therefore, in this paper, the information gain method [33] is successfully applied to investigate the best performance of the proposed model in terms

of accuracy and computing time. Furthermore, we consider a loop in which features are continuously (one by one) added to the prediction model based on the highest-ranked. As a result, we select only 60 highest-ranked pixels out of 784 features. The process of reducing the 784 features or pixels is to dispense with a set of pixels that do not represent hand gestures and, thus, reduce the mathematical complexity of the feature space and enable the modeling algorithm to work more efficiently, as shown in the results.

### 4.1.2. Hand Gesture Classifiers

After selecting the best 60 pixels, the dataset is divided into a training dataset of 66% and a testing dataset of 33%. In this paper, we consider four well-known classifiers known as MLP [34], SVM [35], RFT [36], and Ada-Boost [37] using orange data mining software [38] to classify 24 classes of hand motions. In addition, we use 5-fold cross-validation for all classifiers. We choose 5-fold cross-validation because this is usually accurate compared to other k using different experiments. Table 1 lists the key parameters of each classifier. At the end of this model, all classifiers are combined using the voting process to enhance the performance of the proposed model by selecting the best classifier through high evaluation metrics.

**Table 1.** Key parameters of the hand gesture learning process.

| Classifiers | Key Parameters |
| --- | --- |
| ANN | Maximum number of iterations: 200<br>Neurons in hidden layers: 100<br>Activation function: ReLu<br>Solver: Adam<br>Regularization: 0.0001 |
| SVM | Iteration limit: 100<br>Cost value: 1<br>Kernel: RBF<br>Regression loss epsilon ($\varepsilon$): 0.10<br>Numerical tolerance: 0.0010 |
| RFT | Number of trees: 10<br>Number of attributes at each split: 5<br>Individual tree depth limit: 3<br>Do not split subset smaller than: 5 |
| Ada-Boost | Number of estimators: 50<br>Classification algorithm: SAMME.R<br>Base estimator: Tree<br>Learning rate: 1<br>Regression loss function: Linear |

### 4.1.3. Hand Gesture Recognition (Sign Prediction)

The new gesture is consecutively fed through the prediction model to recognize the current gesture. After that, the features of the recognized gesture are used as input to perform the new password generation. The only drawback of this method is that someone looks at the user who is entering the password using hand gestures. In other words, the intruder learns how the user enters the hand gestures in terms of the type and number of signs. To overcome this problem, we propose to use an extra variable combination of a hand gesture with at least four signs to increase complexity. Herein, we use the same traditional mechanism of creating a new password by prompting users to sign their emotions twice to verify the correct matching between the same hand gestures. For example, if the user wants to enter four signs (e.g., signs A, K, L, and F), this user must repeat these signs twice. In other words, users will be asked to confirm their passwords by signing the sequence (A, K, L, and F) twice, one for the first sequence (new password) and the other for the second sequence (confirm password). If the new password does not match the confirmed

password, this means that users are missing their sequence of hand gestures or the hand gestures are incorrect.

### 4.1.4. Proposed Password Generation

Initially, users apply their sense of randomness when generating a password. This randomness is about the variety of choosing several hand motions (e.g., a user randomly chooses at least four hand motions out of 24 labels) around what they think is private. In contrast, the proposed password generator tries to generate a strong and memorable password depending on the label type for each user's hand motion and similar features to predicted motion. These features are retrieved using Table 2, which considers a snapshot of the training MNIST images dataset. For example, if the users choose four sign languages, such as 1, 2, 3, and 4, then the features of these motions are highlighted in the following table. Thus, the retrieved 4*60 features are used to generate a new password.

**Table 2.** A snapshot of the training MNIST images dataset.

| Label | F1 | F2 | . . . | F60 |
|:---:|:---:|:---:|:---:|:---:|
| 0 | 134 | 119 | . . . | 120 |
| 1 | 224 | 168 | . . . | 228 |
| 2 | 118 | 109 | . . . | 130 |
| 3 | 178 | 135 | . . . | 181 |
| 4 | 212 | 198 | . . . | 197 |
| 5 | 145 | 143 | . . . | 141 |
| 6 | 63 | 111 | . . . | 60 |
| 7 | 110 | 111 | . . . | 113 |
| 8 | 202 | 188 | . . . | 187 |
| 10 | 112 | 125 | . . . | 112 |
| 11 | 43 | 71 | . . . | 59 |
| 12 | 210 | 222 | . . . | 200 |
| 13 | 255 | 238 | . . . | 255 |
| 14 | 63 | 53 | . . . | 70 |
| 15 | 103 | 101 | . . . | 118 |
| 16 | 78 | 87 | . . . | 75 |
| 17 | 172 | 211 | . . . | 168 |
| 18 | 194 | 196 | . . . | 204 |
| 19 | 168 | 60 | . . . | 195 |
| 20 | 94 | 144 | . . . | 156 |
| 21 | 161 | 138 | . . . | 171 |
| 22 | 115 | 173 | . . . | 112 |
| 23 | 99 | 83 | . . . | 108 |
| 24 | 90 | 100 | . . . | 87 |

The input code (180 decimal digits equal 1440 bits binary) is produced as input to the generator by applying the XORing operator among the selected sequential user's sign motions. The input code is divided into two parts. One is equal to 80% (144 decimal digits) of the code as input, and the other is equal to 20% (36 decimal digits) as the key to the HMAC hash message authentication code. Hash 256 produces 256 bits or 32 blocks of hexadecimal, dividing 32 blocks into two blocks; each one has 16 blocks. For each round of the proposed algorithm, one block of 8 bits is taken and divided into left (L) and right (R) parts; each one has 4 bits. To produce a new right (R) part using XOR operation between (L) and (R), as well as being given a new left (L) by using the proposed S-Box substitution box to select 4 new bits, the content of the S-Box is one digit of hexadecimal (4 bits). One ASCII character is produced for each round and is concatenated to the password string. This process applies to the second 16 blocks to produce a second password. All steps are shown in Figure 5.

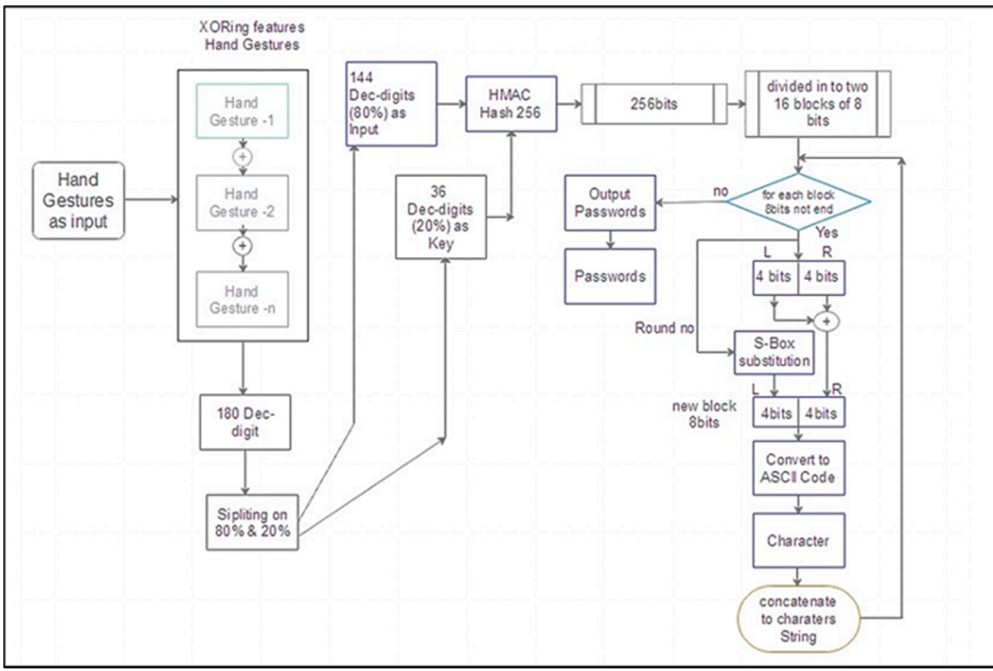

**Figure 5.** The proposed password generator.

The proposed S-Box is a vital technique widely used to produce new values using a specific table, as shown in Figure 6. The value inside the S-Box is a hexadecimal digit selected from a cross between two binary numbers of a row and two binary digits of a column. The input of the S-Box is the number of a round of generators, and the output is the hexadecimal digit of 4 bits that represents the new left half of 8 bits in the generator's steps. For example, the value of 0010 is the sample input value to the S-Box that indicates that the mechanism of the proposed S-Box is working, and the first two digits (00) represent the row value. The second two digits (10) represent the column value, and the output of a crossing row and column is (6) as a hexadecimal value.

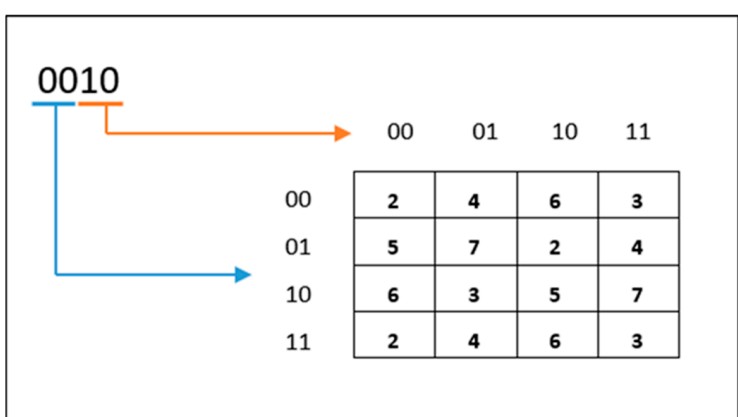

**Figure 6.** The proposed S-Box.

### 4.2. The Proposed Password Strength Estimation Using Ensemble Learning

After generating new passwords using the password generation process, these passwords are passed to this model to check the strength of passwords against external attacks. This proposed model can be defined into four main processes: handling imbalanced datasets, feature extraction, password strength classifiers, and strength estimation, as shown in Figure 7.

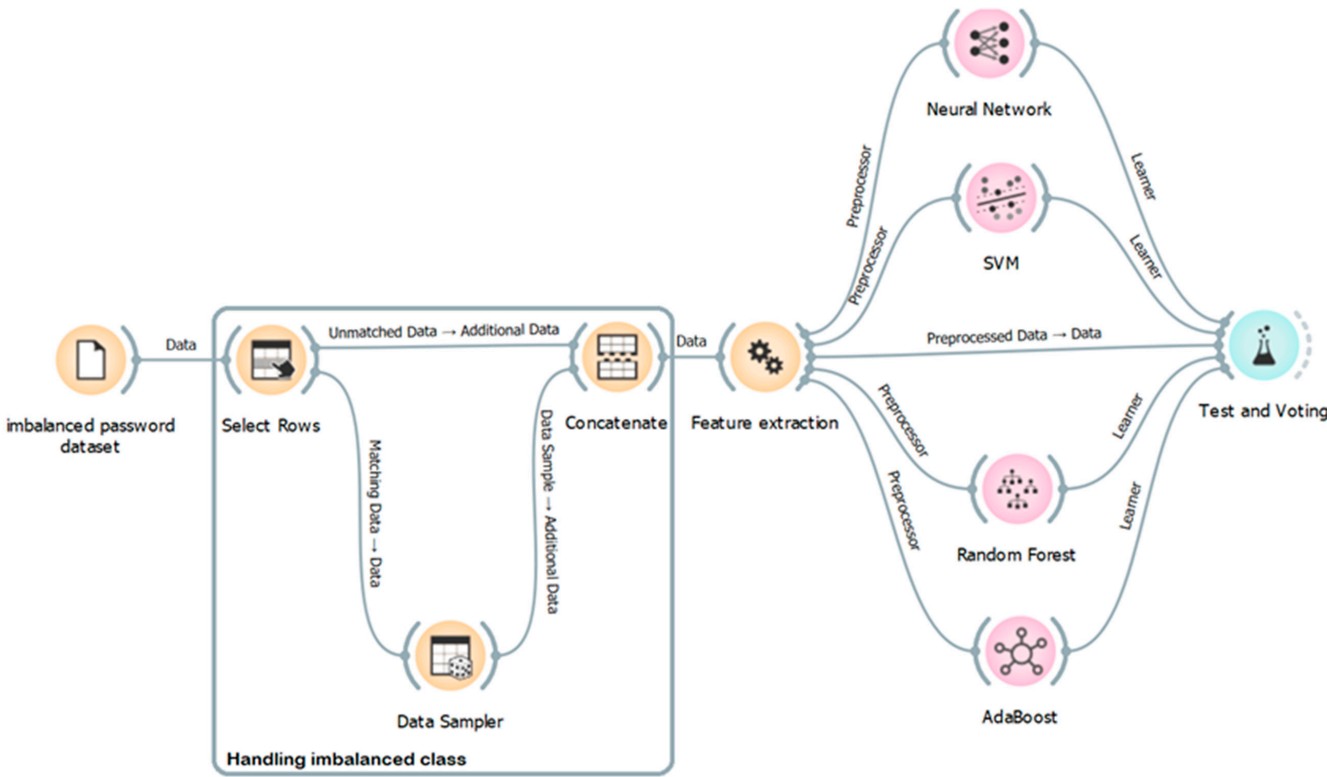

**Figure 7.** Architecture of password strength classification using ensemble learning.

#### 4.2.1. Handling Imbalanced Dataset

As mentioned above, the strength class is imbalanced. To handle this problem in this paper, the data's majority class points (strong value) are resampled to make them equal to the minority class points (weak and medium values) via the under-sampling method. We use the select rows tool to separate the minority class (strong value) and connect it to the data sampler tool, where we choose both fixed sample sizes = 8951. After under sampling, the percentage distributions of class values are 34.22%, 34.22%, and 31.56% for weak, strong, and medium, respectively.

#### 4.2.2. Proposed Feature Extraction Method

In this paper, seven features are extracted from the password strength dataset, such as password length, number of uppercase characters, number of lowercase characters, number of digits, and number of special characters, diversity, and entropy, as shown in Table 3. Herein, we briefly illustrate two of the proposed features as follows:

**Table 3.** A snapshot of the training password strength dataset.

| Id. | Password | Length | No. of Uppercase | No. of Lowercase | No. of Digits | No. of Special Chr. | Diversity | Entropy | Strength |
|-----|----------|--------|------------------|------------------|---------------|---------------------|-----------|---------|----------|
| 1. | p2share | 7 | 0 | 6 | 1 | 0 | 3 | 2.8074 | 0 |
| 2. | j09000 | 6 | 0 | 1 | 5 | 0 | 2 | 1.2516 | 0 |
| 3. | 5gzj5uf | 7 | 0 | 5 | 2 | 0 | 4 | 2.5216 | 0 |
| 4. | winxp; | 6 | 0 | 5 | 0 | 1 | 2 | 2.585 | 0 |
| 5. | ZM9199 | 6 | 2 | 0 | 4 | 0 | 2 | 1.7925 | 0 |
| 6. | kzde5577 | 8 | 0 | 4 | 4 | 0 | 2 | 2.5 | 1 |
| 7. | YADHJIGSAWS11 | 13 | 11 | 0 | 2 | 0 | 2 | 3.2389 | 1 |
| 8. | khurram_ | 8 | 0 | 7 | 0 | 1 | 2 | 2.75 | 1 |
| 9. | AS0130066 | 9 | 2 | 0 | 7 | 0 | 2 | 2.4194 | 1 |
| 10. | 123_456_789 | 11 | 0 | 0 | 9 | 2 | 5 | 3.2776 | 1 |
| 11. | !"64~J"bL+^/NGZ$CNfUbE)?Pvapt9 | 30 | 10 | 7 | 3 | 10 | 19 | 4.7069 | 2 |
| 12. | 1q2w3e4r5t6y7u8i9o0P | 20 | 1 | 9 | 10 | 0 | 20 | 4.3219 | 2 |
| 13. | 248sUqiFEJuRag | 14 | 5 | 6 | 3 | 0 | 8 | 3.8074 | 2 |
| 14. | 678CuLeJAPazob | 14 | 5 | 6 | 3 | 0 | 7 | 3.8074 | 2 |
| 15. | Me&ren102003000 | 15 | 1 | 4 | 9 | 1 | 5 | 2.7396 | 2 |

Password diversity: This feature returns how many sequences of symbols appear in the password and differ from adjacent symbols. For example, if the user password is shell32dll, the diversity is three because this password contains only three sequences such as "shell", "32", and "dll".

Password entropy: Password entropy estimates how complex a certain password would be to guess. Password entropy is determined by the sequence of characters, which can be increased by using lowercase, uppercase, digits, and symbols, as well as the length of the password. The proposed entropy returns the amount of information about characters occurring in a password. In other words, it counts the number of occurrences of every character using the following equation.

$$s = -\sum_{i=1}^{w} p \log_2 p \tag{1}$$

where $w$ is the total number of instances and $p$ is the probability that any instance $i$ occurs in the password. For example, consider the password as j09000.

For this password, the value of $w = 3$, because there are three instances (j, 0, and 9), one instance of "j", four instances of "0", and one instance of "9"; then, the value of entropy is calculated as follows:

$$s = -\frac{1}{6}log\frac{1}{6} - \frac{4}{6}log\frac{4}{6} - \frac{1}{6}log\frac{1}{6} = 1.2516 \tag{2}$$

### 4.2.3. Password Strength Classifiers

This process begins after the feature extraction process, where each password is classified according to its weak, medium, or strong strength. Fortunately, training a password's strength is similar to training a user's hand gestures to choose the same classifiers (MLP, SVM, RFT, and Ada-Boost) and the parameters of each classifier. The process of selecting these classifiers depends on the performance of each one in terms of AUC, Accuracy, F1-measures, Precision, and Recall.

### 4.2.4. Password Strength Estimation (Prediction)

The new password is also passed to the prediction model to estimate its strength. If the new password's strength is either medium or weak, the proposed model will reject it and generate a new password. Otherwise, it will accept a new password. The life cycle of the proposed model starts with creating a password and ends with estimating the password until we obtain a strong or very strong password.

## 5. Results and Discussion

### 5.1. The Proposed Password Strength Estimation Using Ensemble Learning

Conventionally, all extracted features from the MNIST dataset (784 total pixels per hand motion) were fed to the proposed ensemble learning (ANN, SVM, RFT, and AdaBoost). Fivefold cross-validation was used in the proposed classifier models with a training size of 66% and a testing size of 33%. All experimental results in Figure 8 showed that ANN, SVM, and RFT had superior performance on the MNIST dataset than AdaBoost. ANN and SVM classifiers obtained an AUC, Accuracy, F1-measures, Precision, and Recall of 100%, while RFT obtained an AUC, Accuracy, F1-measures, Precision, and Recall of 99.90%, 98.21%, 98.21%, 98.21%, and 98.21%, respectively. This paper applied information-gain-based feature selection to select the best features for each sample, giving higher accuracy. The information gain method gave 60 highest-ranked pixels out of 784 features. These selected features were fed to the same proposed ensemble learning. All experimental results in Figure 9 showed that ANN and RFT had superior performance on the MNIST dataset than other classifiers. ANN classifier obtained an AUC, Accuracy, F1-measures, Precision, and Recall of 100%, 99.97%, 99.97%, 99.97%, and 99.97, respectively, while RFT obtained an AUC, Accuracy, F1-measures, Precision, and Recall of 99.68%, 94.77%, 94.77%, 94.78%,

and 94.77%, respectively. As a result, we actually obtained a very slight decrease in all five mercies. However, that could make the training time prohibitively short. Table 4 presents experimental results for the MNIST dataset based on our ablation study using 60 highest-ranked pixels from 20 to 120 iterations for both Neural Network and SVM.

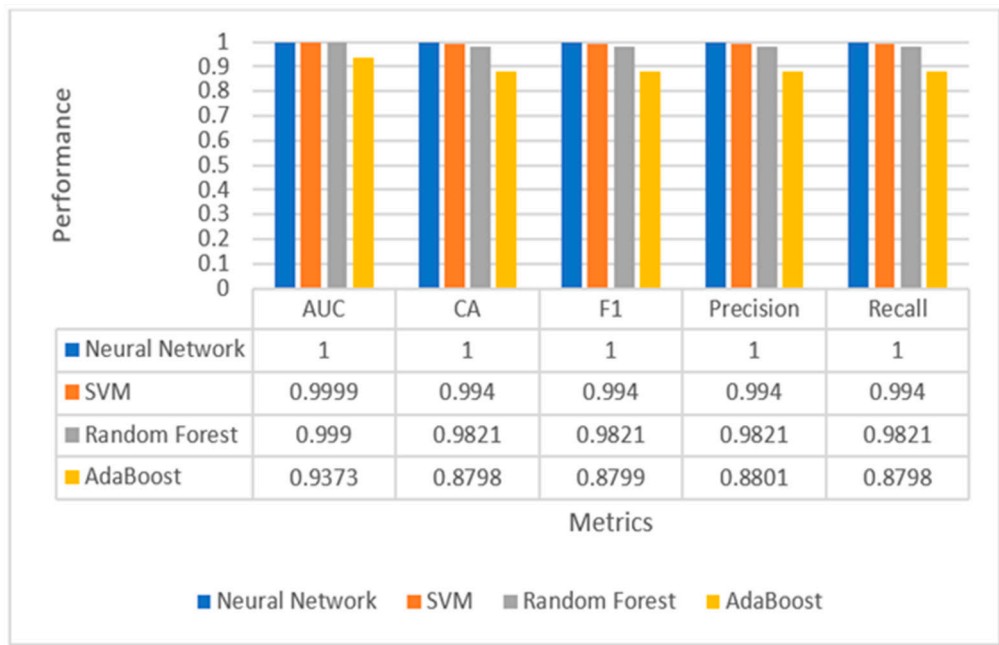

**Figure 8.** Performance evaluation of multiple classifiers for hand gesture recognition using all features.

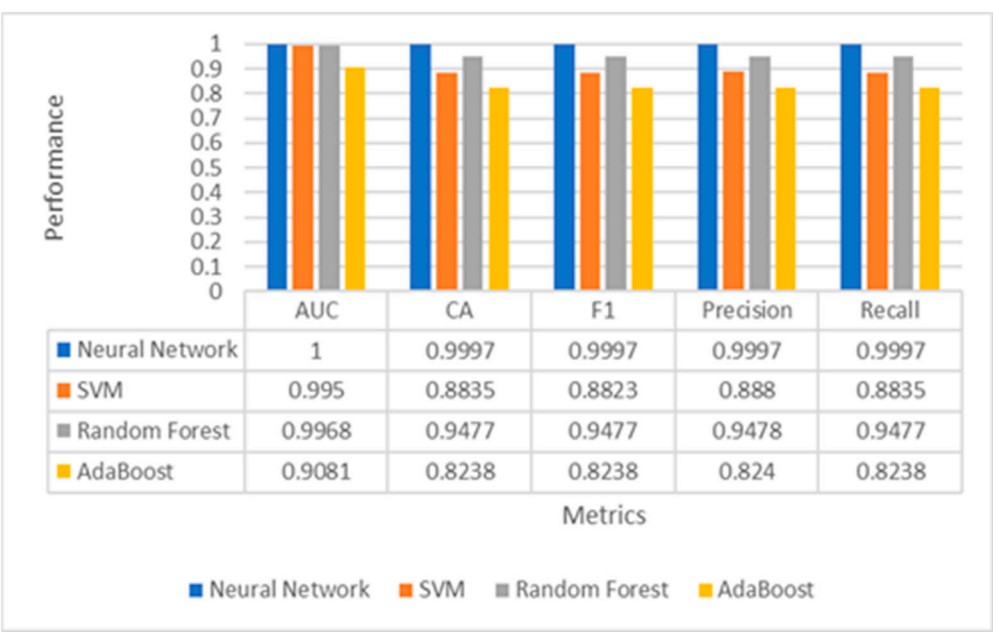

**Figure 9.** Performance evaluation of multiple classifiers for hand gesture recognition using only 60 features.

### 5.2. Performance Comparison of Hand Gesture Recognition with State of the Art

This section analyzes the proposed ensemble learning method and compares the results to other recent techniques. Compared to all baseline techniques, the proposed strategy performed exceptionally well. Through empirical results and comparisons with state-of-the-art, we show that machine learning shines over deep learning when it comes to problems that are not as complex as hand gesture classification.

**Table 4.** Ablation study results on the MNIST dataset.

| Iteration | SVM | | | | | MLP | | | | |
|---|---|---|---|---|---|---|---|---|---|---|
| | Train Time (s) | Test Time (s) | AUC | CA | F1 | Train Time (s) | Test Time (s) | AUC | CA | F1 |
| 20 | 97.020 | 34.977 | 0.9772 | 0.6763 | 0.6739 | 44.769 | 0.567 | 1 | 0.9986 | 0.9986 |
| 40 | 171.315 | 50.169 | 0.9858 | 0.7671 | 0.7641 | 86.629 | 0.506 | 1 | 0.9995 | 0.9995 |
| 60 | 251.76 | 61.165 | 0.9858 | 0.7671 | 0.7641 | 124.953 | 0.607 | 1 | 0.9995 | 0.9995 |
| 80 | 360.076 | 77.605 | 0.9921 | 0.8362 | 0.833 | 178.028 | 0.533 | 1 | 0.9995 | 0.9995 |
| 100 | 365.26 | 76.316 | 0.9948 | 0.8835 | 0.8823 | 212.49 | 0.516 | 1 | 0.9996 | 0.9996 |
| 120 | 445.276 | 90.723 | 0.9948 | 0.8835 | 0.8823 | 216.282 | 0.516 | 1 | 0.9996 | 0.9996 |

Table 5 shows that the proposed ensemble learning accuracy was up to 100% without using the feature selection method. At the same time, it reached a minimal decrease to 99.97 when using only 60 features. Thus, we increased the model accuracy and decreased the model training time.

**Table 5.** Performance comparison of the proposed hand gesture recognition with state-of-the-art.

| Ref. | Gesture Type | Accuracy (%) |
|---|---|---|
| Jalal, M.A. et al. | 24 ASL gestures | 99.00 |
| Chong, T.-W. et al. | 26 ASL gestures (A–Z) and 36 ASL gestures (A–Z, 0–9) | 93.81 |
| Aly, W. | 24 ASL | 88.70 |
| et al. | 26 English alphabets | 94.34 |
| Das, P. et al. | 26 English alphabets | 95.18 |
| Alon, H.D. et al. | 10 signs of 0 to 9 digits | 87.50 |
| Chavan, S. et al. | 24 ASL gestures | 99.67 |
| our proposed method | 24 ASL gestures | 99.97 |
| our proposed method | 24 ASL gestures | 100.00 |

*5.3. Performance Assessment of Multiple Classifiers for Password Strength Estimation*

We trained this model in the same conditions in which hand gesture classification was trained. The reason was due to the high performance of these four classifiers. The experimental results in Figure 10 showed that all classifiers (ANN, SVM, RFT, and AdaBoost) had superior performance using a balanced password strength dataset. AUC, Accuracy, F1-measures, Precision, and Recall reached 100% due to our data balancing process. Our experiments conclude that our proposed system is sufficient for detecting and analyzing the strength of passwords.

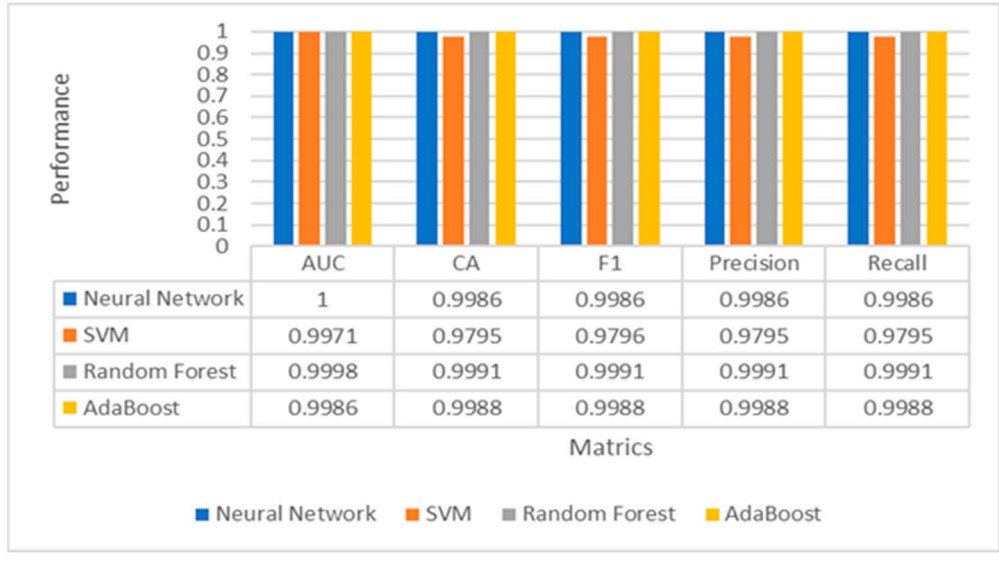

**Figure 10.** Performance evaluation of multiple classifiers for password strength estimation.

### 5.4. Performance Comparison of Proposed Password Strength Estimation with State of the Art

This section also analyzes the proposed ensemble learning method and compares the results to those of other recent techniques. The results of the proposed approach were compared with the outstanding classical classifiers such as Decision tree (DT), Naive Bayes (NB), Logistic Regression (LR), ANN, and RFT [20,21]. Table 6 summarizes the experimental results, highlighting the best performance metrics values. The Accuracy, Precision, and Recall of our classifiers reached 99%. Thus, an ensemble's performance is superior to those of the other classifiers.

**Table 6.** Performance comparison of the proposed password strength estimation with state-of-the-art.

| Ref. | Classifier | Accuracy (%) | Precision (%) | Recall (%) |
|------|-----------|--------------|---------------|------------|
| Farooq, U. [21] | DT | 99 | 98 | 97 |
| | NB | 87 | 78 | 81 |
| | LR | 89 | 81 | 84 |
| | RFT | 95 | 94 | 91 |
| | ANN | 92 | 89 | 87 |
| Rathi, R. et al. [20] | ANN | 77 | - | - |
| | LR | 81 | - | - |
| our proposed method | ANN | **99** | **99** | **99** |
| | SVM | **97** | **97** | **97** |
| | RFT | **99** | **99** | **99** |
| | AdaBoost | **99** | **99** | **99** |

### 5.5. The Analysis of Proposed Password Generation and Strength Estimation

There were six critical steps in the implementation process. First, we ran an experiment asking ten users to choose at least four hand motions. For example, the first user chose four motions, as shown in Table 5. Second, each of these four motions was classified using the hand gesture recognition model to distinguish the type or label of hand motions. As a result, these four hand motions were classified into 1, 2, 3, and 4. Third, we used Table 2 to retrieve 60 features for each of hand motion. Fourth, the password generator model was applied to generate two strong and memorable passwords using 60 features of hand motion as follows:

Password 1 = ,Gc6X~$Cd2Wv*Gk8

Password 2 = Zp'@d8]u'Na;W}-F

Fifth, after password generation, the features of these two passwords were extracted using the feature extracted model. As shown in Table 7, these extracted features were 22, 5, 4, 3, 10, 16, and 4 (password length, number of uppercase, number of lowercase, number of digits, number of special characters, password diversity, and password entropy, respectively). Finally, the password estimation model was used to estimate the strength of each generated password. Table 8 shows a sample of the passwords examined and their strengths as determined by existing password meters and the proposed model. These passwords were evaluated against different password meters (password strength checker, password meter, password monster, and proposed model). In addition, we tested our Ensemble model with newly generated passwords. The result illustrated in Table 8 shows that 100% of the passwords were classified as very strong. From these comparisons, it can be inferred that our proposed strength estimator guessed that most of the generated passwords were very strong compared to the existing estimators that were also very strong. Furthermore, it can be observed that there was no significant difference between the evaluations, due to the strength of the characteristics of all generated passwords, such as length, diversity, and entropy.

**Table 7.** The features of the generated passwords.

| Hand Gestures Selection | Password | Length | No. of Uppercase | No. of Lowercase | No. of Digits | No. of Special Chr. | Diversity | Entropy |
|---|---|---|---|---|---|---|---|---|
| 1,2,3,4 | ,Gc6X~$Cd2Wv*Gk8 | 22 | 5 | 4 | 3 | 10 | 16 | 4 |
| | Zp'@d8]u'Na;W}-F | 16 | 4 | 4 | 1 | 7 | 14 | 4 |
| 10,12,14,16 | 'Cc<Xp!He=Rz$Lk> | 22 | 5 | 5 | 0 | 12 | 16 | 4 |
| | Qr&Mn6_z!Do4Yp.C | 16 | 5 | 5 | 2 | 4 | 16 | 4 |
| 20,21,22,23 | /Om>\p#@a2Vt'Mm2 | 19 | 3 | 5 | 2 | 9 | 14 | 4 |
| | ^q-Ab2P\|$Ch>Wv!E | 19 | 5 | 4 | 1 | 9 | 15 | 4 |
| 1,11,17,19 | .Eg7T{$Fd3Tr/Ln4 | 22 | 5 | 4 | 3 | 10 | 16 | 4 |
| | Yv$Ah?Xu(Bd;Xw#M | 16 | 6 | 5 | 0 | 5 | 16 | 4 |
| 24,2,6,7,8,9 | *Fk3Yv)Ca9Zp+Gc< | 22 | 5 | 5 | 2 | 10 | 16 | 4 |
| | Q}.Bf2Px$Eo8[s#O | 16 | 5 | 4 | 2 | 5 | 15 | 4 |
| 20,24,19,3,6 | #Il:Uy%@m=Q}.@h | 21 | 3 | 4 | 0 | 14 | 13 | 4 |
| | 9Pw)@b<S} Gj7Yu'B | 23 | 5 | 4 | 2 | 12 | 17 | 4 |
| 2,20,15,5 | &Hd0Yy(Ij9Sq,Oj9 | 19 | 5 | 5 | 3 | 6 | 17 | 4 |
| | Pp#F'1Wu&Kg>]{'H | 16 | 5 | 3 | 1 | 7 | 13 | 4 |
| 12,13,14,15 | 'Db4Sw.Lo9^s*Mc4 | 22 | 4 | 5 | 3 | 10 | 17 | 4 |
| | \q/Go3\}/Dg4]w&B | 22 | 3 | 4 | 2 | 13 | 15 | 4 |
| 21,4,3,6,8 | Ak=SwOd1Wx#Ac6V} | 19 | 6 | 5 | 2 | 6 | 16 | 4 |
| | *Ea<Ys*F'3_q(B | 20 | 4 | 3 | 1 | 12 | 15 | 4 |
| 7,15,16,22,23,17 | &Le4Xu#Oi:P\|%Fi: | 19 | 5 | 4 | 1 | 9 | 15 | 4 |
| | [z-Jk=_p(@a0\r"K | 19 | 2 | 5 | 1 | 11 | 14 | 4 |

**Table 8.** The estimation results of generated passwords using different password meters.

| Hand Gestures Selection | Password | Password Strength Checker | Password Meter | Password Monster | Proposed Model |
|---|---|---|---|---|---|
| 1,2,3,4 | ,Gc6X~$Cd2Wv*Gk8 | Very Strong | Very Strong | Very Strong | Very Strong |
|  | Zp'@d8]u'Na;W}-F | Very Strong | Very Strong | Very Strong | Very Strong |
| 10,12,14,16 | 'Cc<Xp!He=Rz$Lk> | Very Strong | Very Strong | Very Strong | Very Strong |
|  | Qr&Mn6_z!Do4Yp.C | Very Strong | Very Strong | Very Strong | Very Strong |
| 20,21,22,23 | /Om>\p#@a2Vt'Mm2 | Very Strong | Very Strong | Very Strong | Very Strong |
|  | ˆq-Ab2P∣$Ch>Wv!E | Very Strong | Very Strong | Very Strong | Very Strong |
| 1,11,17,19 | .Eg7T{$Fd3Tr/Ln4 | Very Strong | Very Strong | Very Strong | Very Strong |
|  | Yv$Ah?Xu(Bd;Xw#M | Very Strong | Very Strong | Very Strong | Very Strong |
| 24,2,6,7,8,9 | *Fk3Yv)Ca9Zp+Gc< | Very Strong | Very Strong | Very Strong | Very Strong |
|  | Q}.Bf2Px$Eo8[s#O | Very Strong | Very Strong | Very Strong | Very Strong |
| 20,24,19,3,6 | #Il:Uy%@m=Q}.@h | Very Strong | Very Strong | Very Strong | Very Strong |
|  | 9Pw)@b<S} Gj7Yu'B | Very Strong | Very Strong | Very Strong | Very Strong |
| 2,20,15,5 | &Hd0Yy(Ij9Sq,Oj9 | Very Strong | Very Strong | Very Strong | Very Strong |
|  | Pp#F'1Wu&Kg>]{'H | Very Strong | Very Strong | Very Strong | Very Strong |
| 12,13,14,15 | 'Db4Sw.Lo9ˆs*Mc4 | Very Strong | Very Strong | Very Strong | Very Strong |
|  | \q/Go3\}/Dg4]w&B | Very Strong | Very Strong | Very Strong | Very Strong |
| 21,4,3,6,8 | Ak=SwOd1Wx#Ac6V} | Very Strong | Very Strong | Very Strong | Very Strong |
|  | *Ea<Ys*F'3_q(B | Very Strong | Very Strong | Very Strong | Very Strong |
| 7,15,16,22,23,17 | &Le4Xu#Oi:P∣%Fi: | Very Strong | Very Strong | Very Strong | Very Strong |
|  | [z-Jk=_p(@a0\r"K | Very Strong | Very Strong | Very Strong | Very Strong |

## 6. Conclusions

This paper presents a new intelligent security model for password generation and estimation based on hand gestures. The proposed model allows users to select any type and number of multi-sign-language motions to design strong, memorable passwords and prevent intruders from identifying these motions easily. The experimental results reveal that our proposed strength estimator estimates that all generated passwords are very strong. The key aspects of a very strong password are length, a mix of letters (upper and lower case), symbols, numbers, diversity, entropy, no ties to personal information, and no dictionary words. These key aspects are created from different features of multi-sign-language motions and the S-Box mechanism. In addition, the proposed password generator generates memorable passwords that are easy to remember due to the use of multi-sign language motions. In future work, we will suggest promising ideas that can increase the efficiency of the existing password generator using ensemble deep learning applied to the extended MNIST images dataset.

**Author Contributions:** Conceptualization, A.R.A.; methodology, A.R.A. and B.S.M.; software, A.R.A. and B.S.M.; validation, A.R.A., B.S.M. and M.J.H.; formal analysis, A.R.A. and B.S.M.; investigation, A.R.A.; resources, A.R.A.; data curation, A.R.A.; writing—original draft preparation, A.R.A.; writing—review and editing, A.R.A., B.S.M. and M.J.H.; visualization, A.R.A., B.S.M. and M.J.H.; supervision, A.R.A.; project administration, A.R.A.; funding acquisition, A.R.A., B.S.M. and M.J.H. All authors have read and agreed to the published version of the manuscript.

**Funding:** This research received no external funding.

**Institutional Review Board Statement:** Not applicable.

**Informed Consent Statement:** Not applicable.

**Data Availability Statement:** Data supporting reported results can be found at: https://www.kaggle.com/datasets/datamunge/sign-language-mnist/ (accessed on 1 January 2021). https://www.kaggle.com/datasets/bhavikbb/password-strength-classifier-dataset/ (accessed on 1 January 2022).

**Conflicts of Interest:** The authors declare no conflict of interest.

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
