# Peer review of "Intelligent Security Model for Password Generation and Estimation Using Hand Gesture Features"

_2504-2289, doi:10.3390/bdcc6040116_

Round 1

Reviewer 1 Report

Reviewer-1: This paper presents an intelligent security model for password generation and estimation to address 13 these problems using the ensemble learning approach and hand gesture features.

The proposed architecture is original, well presented and validated by experimental results..
- This paper proposes a method of password generation using hand gesture features and adopt ensemble methods.
- The proposed method outperformed the state-of-the-art methods

Some questions:

1.    Ablation study is missing, i.e., how many hidden layers are quipped and how did the authors decide on the maximum number of iterations for ANN and SVM in Table 1, An ablation study is required to support these experiments.

2.    How do repeat the signs to verify correct matching, does it require repeating each sign twice or repeating the sequences of signs after completing one? Ref. Section 4.1.3

3.    Please re-check the arrows for 00 and 01?

4.    Need to improve the quality of figures 6 and 7 to see more clearly.

5.    In table 4, 5: Add proposed methods or author names corresponding to each reference and prefer to write “our proposed method” for authors’ newly proposed methods in this paper

Add the reference of the dataset in Fig.1

Author Response

Dear Reviewer

Thank you for giving me the opportunity to submit a revised draft of my manuscript titled “Intelligent Security Model for Password Generation and Esti-mation Using Hand Gesture Features” to “Big Data and Cognitive Computing”. We appreciate the time and effort that you and the reviewers have dedicated to providing your valuable feedback on my manuscript. We are grateful to the reviewers for their insightful comments on my paper. We have been able to incorporate changes to reflect most of the suggestions provided by the reviewers. We have highlighted the changes within the manuscript.

Comments from Reviewer 1:

Comment 1: Ablation study is missing, i.e., how many hidden layers are quipped and how did the authors decide on the maximum number of iterations for ANN and SVM in Table 1, An ablation study is required to support these experiments.

Response:  Thank you for pointing this out. We agree with this comment. Therefore, we have done and added ablation study to support our  experiments in page 12 and 13 lines 351-353 as shown in Table 4.

Comment 2: ‎ How do repeat the signs to verify correct matching, does it require repeating each sign twice or repeating the sequences of signs after completing one? Ref. Section 4.1.3

Response:  Thank you for pointing this out. More explanation has been added in page 7 line 241-245 Section 4.1.3.

Comment 3: Please re-check the arrows for 00 and 01?

Response: Thank you for pointing this out. More explanation has been added in page 9‏‎ ‎lines ‎‏274‏‎-278.

Comment 4: Need to improve the quality of figures 6 and 7 to see more clearly.

Response:  Thank you for pointing this out. We agree with this comment.  We improve the quality of figures 6 and 7  in page 9 and 10.

Comment 5: In table 4, 5: Add proposed methods or author names corresponding to each reference and prefer to write “our proposed method” for authors’ newly proposed methods in this paper. Add the reference of the dataset in Fig.1

Response: Thank you for pointing this out. We agree with this comment and some changes have been made.

Reviewer 2 Report

Q1.                  Reduction of features in the dataset may create error while proposing the method. Explain also the reason for reducing features.

Q2.                  What are the main disadvantages of ensemble learning approach?

Q3.                  Specify the sampling period.

Q4.                  The reason for 5-fold cross-validation is required to explanation.

Q5.                  Explain the necessity of counting the the number of occurrences of every character.                      

Author Response

Dear Reviewer

Thank you for giving me the opportunity to submit a revised draft of my manuscript titled “Intelligent Security Model for Password Generation and Esti-mation Using Hand Gesture Features” to “Big Data and Cognitive Computing”. We appreciate the time and effort that you and the reviewers have dedicated to providing your valuable feedback on my manuscript. We are grateful to the reviewers for their insightful comments on my paper. We have been able to incorporate changes to reflect most of the suggestions provided by the reviewers. We have highlighted the changes within the manuscript.

Comments from Reviewer 2:

Comment 1: Reduction of features in the dataset may create error while proposing the method. Explain also the reason for reducing features..

Response:  Thank you for pointing this out. In fact, the process of reducing the 784 features or pixels is to dispense with a set of pixels that do not represent hand gestures and thus reduce the mathematical complexity of the feature space and enable the modeling algorithm to work more efficiently as shown in the results. More explanation has been added on page 6 lines 213-216.

Comment 2: ‎ What are the main disadvantages of ensemble learning approach?

Response:  Thank you for pointing this out. There are two main reasons to apply an ensemble model over a single model. Performance: Compared to a single contributing model, an ensemble can predict events more accurately. Generalization: The ensemble model's capability to adapt to new datasets. More explanation has been added on page 5 lines 199-201.  

Comment 3: Specify the sampling period.

Response:  Thank you for pointing this out. We explained this in section ‎4.1.2. Hand Gesture Classifiers‎‎

Comment 4: The reason for 5-fold cross-validation is required to explanation.

Response:  Thank you for pointing this out. We choose ‎5-fold cross-validation because ‎this is usually pretty accurate than other k ‎using different experiments. ‎ We agree with this comment more explanation has been added on page 6 line 222.

Comment 5: Explain the necessity of counting the number of occurrences of every character.          

Response: Thank you for pointing this out. More explanation has been added on page 11 lines 205-208.

Reviewer 3 Report

In this paper, the authors proposed an intelligent security model for password generation and estimation to enhance password security by using the ensemble learning approach and hand gesture features. The method includes two intelligent stages: the first is the password generation stage based on the ensemble learning approach and proposed S-Box. The second is the password strength estimation stage, also based on the ensemble learning approach. However, I still have some concerns as follows.

1. Literature review is not sufficient. More technical papers about data security and intelligent algorithms should be reviewed.

2. The contributions in the paper should be enhanced and presented clearly. Compared with existing works, what are the advantages of the methods proposed in this paper?

3. What are the overheads (e.g., system setup, data storage, computing efficiency, etc.) of the proposed scheme? More analysis and discussions should be provided.

4. There are more opportunities for conducting meaningful experiments to comprehensively evaluate the system performance and overheads.

5. More applications about data security and intelligent algorithms should be investigated. For example:

- An efficient learning-based approach to multi-objective route planning in a smart city, IEEE International Conference on Communications.

- A password-based authentication system based on the CAPTCHA AI problem, IEEE Access.

- When urban safety index inference meets location-based data, IEEE Transactions on Mobile Computing.

- Data security and privacy protection issues in cloud computing, International Conference on Computer Science and Electronics Engineering.

- Smartphone-assisted smooth live video broadcast on wearable cameras, IEEE/ACM International Symposium on Quality of Service.

- Research on AI security enhanced encryption algorithm of autonomous IoT systems, Information sciences.

- U-safety: Urban safety analysis in a smart city, IEEE International Conference on Communications.

Author Response

Dear Reviewer

Thank you for giving me the opportunity to submit a revised draft of my manuscript titled “Intelligent Security Model for Password Generation and Esti-mation Using Hand Gesture Features” to “Big Data and Cognitive Computing”. We appreciate the time and effort that you and the reviewers have dedicated to providing your valuable feedback on my manuscript. We are grateful to the reviewers for their insightful comments on my paper. We have been able to incorporate changes to reflect most of the suggestions provided by the reviewers. We have highlighted the changes within the manuscript.

Comments from Reviewer 3:

Comment 1: Literature review is not sufficient. More technical papers about data security and intelligent algorithms should be reviewed.

Response:  Thank you for pointing this out. We have added more technical papers according to your recommendations in page ‎‏3‏‎ lines ‎‏138‏‎-144.

Comment 2: ‎ the contributions in the paper should be enhanced and presented clearly. Compared with existing works, what are the advantages of the methods proposed in this paper?

Response:  Thank you for pointing this out. We agree with this comment. ‎More explanation has been added in page 2 lines 63-66.

Comment 3: What are the overheads (e.g., system setup, data storage, computing efficiency, etc.) of the proposed scheme? More analysis and discussions should be provided.

Response:  Thank you for pointing this out. We agree with this comment. Therefore, we ‎have done and added an ablation study to support our experiments in pages 12-13 ‎lines 350-352 as shown in table 4.‎

Comment 4: There are more opportunities for conducting meaningful experiments to comprehensively evaluate the system performance and overheads.

Response:  Thank you for pointing this out. We agree with this comment and some ‎changes have been made by adding the results for each iteration with training and testing times as shown in table 4.

Comment 5: More applications about data security and intelligent algorithms should be investigated.

Response: Thank you for pointing this out. We have added more technical papers according to your recommendations in page ‎‏3‏‎ lines ‎‏138‏‎-144.

Reviewer 4 Report

The manuscript is well structured and clearly describes the issue. The chapters follow each other. The logical sequence and description of the research is clear. Please check the cited references 11-15 whether they correspond to the context of the article. I would recommend writing a separate paragraph for the novelty of the article.

Author Response

Dear Reviewer

Thank you for giving me the opportunity to submit a revised draft of my manuscript titled “Intelligent Security Model for Password Generation and Esti-mation Using Hand Gesture Features” to “Big Data and Cognitive Computing”. We appreciate the time and effort that you and the reviewers have dedicated to providing your valuable feedback on my manuscript. We are grateful to the reviewers for their insightful comments on my paper. We have been able to incorporate changes to reflect most of the suggestions provided by the reviewers. We have highlighted the changes within the manuscript.

Comments from Reviewer 4:

Comment 1: The manuscript is well structured and clearly describes the issue. The chapters follow each other. The logical sequence and description of the research is clear. Please check the cited references 11-15 whether they correspond to the context of the article. I would recommend writing a separate paragraph for the novelty of the article.

Response:  Thank you for pointing this out.  The mentioned references from 11 to 15 in section 2.2   are related works to the proposed password strength estimation model.  In addition, the ‎main contributions‎ have been changed and added some information about the novelty of the article.

Round 2

Reviewer 3 Report

All problems have been solved. More references and discussion about applications could be added to enhance the imapct of this work.

Author Response

Dear Reviewer

Thank you for giving me the second opportunity to submit a revised draft of my ‎manuscript ‎titled “Intelligent Security Model for Password Generation and ‎Esti‎mation Using Hand ‎‎Gesture Features” to “Big Data and Cognitive ‎Computing”. We again appreciate the time ‎and effort that you to providing your valuable feedback ‎on my manuscript.  ‎Comments from Reviewer 3:‎

Comment 1: More applications about data security and intelligent algorithms ‎should be investigated. ‎

Response: Thank you for pointing this out again. We have added more applications with 9 references using a new paragraph in section 2.3 lines 139-165.
